# miR-126 Decreases Proliferation and Mammosphere Formation of MCF-7 and Predicts Prognosis of ER+ Breast Cancer

**DOI:** 10.3390/diagnostics12030745

**Published:** 2022-03-18

**Authors:** Zahraa S. Msheik, Farah J. Nassar, Ghada Chamandi, Abdul Rahman Itani, Emanuala Gadaleta, Claude Chalala, Nisreen Alwan, Rihab R. Nasr

**Affiliations:** 1Department of Anatomy, Cell Biology and Physiological Sciences, Faculty of Medicine, American University of Beirut, Beirut 11-0236, Lebanon; zsm10@mail.aub.edu (Z.S.M.); gc21@aub.edu.lb (G.C.); abed.s.itani@gmail.com (A.R.I.); 2Department of Internal Medicine, Faculty of Medicine, American University of Beirut, Beirut 11-0236, Lebanon; fjn00@mail.aub.edu; 3Pathophysiology of Breast Cancer Team, INSERM U976, HIPI, Université de Paris, 75010 Paris, France; 4Centre for Molecular Oncology, Barts Cancer Institute, Queen Mary University of London, Charterhouse Square, London EC1M 6BQ, UK; e.gadaleta@qmul.ac.uk (E.G.); c.chelala@qmul.ac.uk (C.C.); 5College of Health Sciences, Abu Dhabi University, Abu Dhabi 59911, United Arab Emirates; nisreen.alwan@adu.ac.ae

**Keywords:** breast cancer, estrogen receptor-positive, miR-126, SLC7A5 (LAT1)

## Abstract

Breast cancer (BC) is a major health burden that affects over one million women each year. It is the most prevalent cancer in women and the number one cancer killer of them worldwide. Of all BC subtypes, estrogen receptor-positive (ER+) BC is the most commonly diagnosed. The objective of this study is to investigate the contribution of miR-126 in the tumorigenesis of ER+ BC. miR-126 was downregulated in ER+ BC tissues from young breast cancer patients, as shown through miRNA microarray analysis and RT-qPCR. Subsequently, the effect of the modulation of miR-126 levels on the proliferation, cell cycle progression, and spheres formation of the ER+ BC cell line, MCF-7, was assessed by MTT assay, PI analysis, and mammosphere formation assay, respectively. miR-126 overexpression significantly decreased MCF-7 proliferation and mammosphere-forming ability, but did not affect cell cycle progression. Then, in silico analysis determined SLC7A5, PLXNB2, CRK, PLK2, SPRED1, and IRS1 as potential targets of miR-126. RT-qPCR data showed that miR-126 overexpression significantly downregulated SLC7A5 and PLXNB2 mRNA levels in MCF-7. Finally, in silico survival analysis showed that high expression of miR-126 or low expression of SLC7A5 correlated with better overall survival (OS) of ER+ BC patients. Overall, our study suggests that miR-126 might play a tumor suppressor role in ER+ BC. miR-126 and SLC7A5 might also be considered potential prognostic biomarkers in ER+ BC.

## 1. Introduction

Breast cancer (BC) is the most frequent cancer and the leading cause of cancer death in women worldwide. In 2018, it was estimated that BC in women accounts for 24.2% of all cancer cases and 15% of all cancer mortalities [1]. BC is a heterogeneous type of cancer composed of different subtypes characterized by distinct clinical outcomes and responses to therapies [2]. Estrogen receptor-positive (ER+) BC, which expresses the estrogen receptor that drives the growth of the tumor in response to their ligand, estrogen, constitutes about 80% of all breast carcinoma. The activation of the transcription factor ERα promotes the transcription of target genes involved in several oncogenic processes, such as epithelial-to-mesenchymal transition [3]. Although it is characterized by the best prognosis and response to endocrine therapy that can block the mitogenic estrogen activity [4,5], some of these tumors develop resistance to the antiestrogens and long-term risk for recurrence [5,6]. Therefore, uncovering new players in ER+ BC development might lead to a better understanding of its tumorigenesis and response to therapies.

microRNA (miRNA), a class of small, endogenous, non-coding RNA molecules, approximately 22 nucleotides in length [7], are shown to be involved in the development of several diseases including cancer [8]. Since they were first discovered in *C. elegans* in 1993, their role has shifted from transcriptional noise to post-transcriptional gene expression modulators [9]. miRNAs modulate gene expression by binding to the 3′-untranslated regions (3′-UTR) of their target mRNAs leading to their degradation or translational repression [7]. As such, they play diverse roles in critical processes such as cell proliferation, migration, and differentiation, therefore contributing to or inhibiting the progression of multiple diseases [1]. Multiple studies have shown that miRNAs are aberrantly expressed in BC and play diverse roles in its tumorigenesis [10]. Interestingly, several miRNAs are shown to be dysregulated in ER+ BC and play a role in ER regulatory mechanisms, thus contributing to ER+ breast carcinogenesis [11].

In 2017, our group conducted a global miRNA microarray analysis to identify specific miRNA signatures in Lebanese BC patients. We reported the significant dysregulation of 173 mature miRNAs in ER+ BC tissues as compared to normal adjacent tissues (NAT) [12]. Of these miRNAs, miR-126 was significantly downregulated, which we were further interested to investigate in BC patients younger than 40 years old. miR-126, or miR-126-3p, is an endothelial-specific miRNA whose aberrant expression was described in several hematological and solid tumors. It is commonly downregulated in most cancers, such as lung, pancreatic, colorectal, esophageal, and other cancers, and shows tumor-suppressive properties [13]. In addition, its downregulation acts as a significant predictor of poor prognosis in different cancers [14,15]. However, in some cases, such as acute myeloid leukemia (AML), where miR-126 levels were reported to be upregulated in patients with the disease, miR-126 enhances cancer progression [16,17]. In breast cancer, miR-126 was reported to be downregulated in BC tissues as compared to normal adjacent ones [18,19]. Interestingly, plasma levels of this miRNA along with three other miRNA (miR-19a, miR-20a and miR-155) were reported to discriminate early-stage BC from metastatic BC patients with an AUC of 0.802 [20]. Low miR-126 expression was also associated with shorter overall survival as compared to higher expression for early BC patients but not in metastatic BC. This was confirmed in a multivariate analysis that showed miR-126 as an independent predictor of shorter OS (HR: 2.558, 95% CI: 1.177–5.560; *p* = 0.018). Another study showed that its low expression was associated with poor metastasis-free survival in BC patients [21]. Hence, this stimulated our interest in further elucidating the role of miR-126 in BC tumorigenesis, specifically in ER+ BC.

In this study, first, we validated the microarray data of miR-126 expression in ER+ BC tissues by RT-qPCR. Then, we identified the effect of overexpressing miR-126 on cell proliferation, cell cycle progression, mammosphere formation, and expression of potential targets. Finally, correlation between the expression of miR-126 or the dysregulated targets and overall survival of ER+ BC patients was assessed.

## 2. Results

### 2.1. Downregulation of miR-126 in ER+ BC Tissues

miRNA microarray analysis for 19 breast tumor tissues versus 5 normal adjacent breast tissues (NAT) collected from young Lebanese ER+ BC revealed a total of 63 differentially expressed miRNAs with a fold change greater than 2 and a *p*-value < 0.05 (Figure 1A). Of these miRNAs, miR-126 was not only significantly downregulated in patients younger than 40 years old (*p* < 0.05) but there was a fold change greater than 2, which is not the case when considering all ages or those over 40 years.

To further validate the dysregulation of miR-126, we performed RT-qPCR on 40 tumors (19 samples < 40 years and 21 samples > 40 years) and 19 NAT. When analyzing miRNA expression in the different age groups as compared to NAT, we found that miR-126 was significantly downregulated in both groups with *p* = 0.001 in patients younger than 40 years and *p* = 0.0002 in patients older than 40 years (Figure 1B).

### 2.2. Overexpression of miR-126 in MCF-7 upon Transfection with miR-126 Mimics

To explore the role of miR-126 in ER+ BC, MCF-7 was transfected with FAM-labeled miR-126 mimic and negative control (NC) duplex. Transfection efficiency was validated by flow cytometry 24 h post transfection (Figure 2). Then, miR-126 levels were further validated in the transfected cells 24 h post transfection by RT-qPCR, with RNU6B as an endogenous control (Appendix A).

### 2.3. Decreased MCF-7 Cell Proliferation and No Effect on Cell Cycle Progression upon Overexpressing miR-126

To determine the effect of miR-126 overexpression on the proliferation of MCF-7, MTT assay was done at 48 and 72 h post transfection. miR-126 significantly decreased the proliferation of MCF-7 at 72 h post transfection (*p* = 0.034) (Figure 3A). Then, to determine the effect of miR-126 on the cell cycle of MCF-7, propidium iodide (PI) assay was performed at 24 and 48 h post transfection. Our data showed that there was no difference in the cell cycle phases of miR-126 mimic–transfected cells when compared to NC duplex–transfected cells at 24 or 48 h post transfection (Figure 3B,C).

### 2.4. Decreased Mammosphere-Forming Ability of MCF-7 upon Overexpressing miR-126

To determine the effect of miR-126 on the sphere formation of MCF-7, a mammosphere formation assay was performed (Figure 4A). Our results showed that there was a significant decrease in the mammosphere-forming efficiency (MFE) of miR-126–transfected MCF-7 (*p* = 0.0301) when compared to NC-transfected cells of the primary generation. To check the effect of miR-126 on the self-renewal of the cells, spheres were propagated. Again, miR-126 was able to reduce the MFE in MCF-7 (*p* = 0.0145) of the secondary generation (Figure 4B).

### 2.5. SLC7A5 and PLXNB2 as Potential Targets of miR-126

miR-126 targets were selected through two predicted target databases, microT-CDS and TargetScanHuman 7.2, and an experimentally validated database Tarbase 7.0. Further selection was performed based on the literature through a PubMed search regarding validated relation of miR-126 with BC and other cancers. As a result, the following targets were selected: SLC7A5, PLXNB2, CRK, PLK2, SPRED1, and IRS1 (Table 1).

To explore whether miR-126 targets SLC7A5, PLXNB2, CRK, PLK2, SPRED1, and IRS1, RT-qPCR was run on miR-126 mimic-transfected cells as compared to NC duplex–transfected cells 24 h post transfection with GAPDH as an internal control. SLC7A5 and PLXNB2 mRNA were significantly downregulated in MCF-7. On the other hand, mRNA of PLK2, CRK, SPRED1, and IRS1 did not show any significant change in their expression (Figure 5). 

### 2.6. Correlation of High Expression of miR-126 or Low Expression of SLC7A5 with Better Overall Survival of ER+ BC Patients

To determine whether miR-126 or the dysregulated targets (SLC7A5 and PLXNB2) could predict prognosis, in silico survival analysis was used. ER+ BC patients were selected in a similar way to our clinical BC tissue samples previously discussed. In addition, the METABRIC database was selected since it includes patients with a long-term follow-up of 94.2 months. A total of 944 ER+ patients were obtained for miR-126 expression data and a total of 720 ER+ patients were obtained for SLC7A5 and PLXNB2 expression data. BC patients with high expression of miR-126 presented a significant association with better overall survival (OS) compared to patients with low expression of miR-126 (HR = 0.61, 95% CI = 0.48–0.78, *p* = 8.1 × 10^−5^) (Figure 6A). Moreover, high expression of SLC7A5 was significantly associated with poor OS (HR = 2.22, 95% CI = 1.58–3.11, *p* = 2.4 × 10^−6^) (Figure 6B). Interestingly, there was no significant difference in the OS of ER- BC patients between the high and low miR-126 or SLC7A5 expressing groups (Appendix A). On the other hand, PLXNB2 showed no prognostic value as there was no significant difference in the OS of the high- and the low-expressing groups (*p* = 0.46, Appendix A).

## 3. Discussion

Breast cancer is the deadliest type of cancer in women worldwide and in Lebanon. Of all breast cancer subtypes, ER+ BC is the most common subtype diagnosed today. Although genetic predispositions, particularly mutations in *BRCA1* and *BRCA2*, are important drivers of this malignancy, epigenetic alterations may play an important role in the development of this disease [2]. Over the past several years, miRNAs have been found to play diverse roles in various types of cancer, including breast cancer [22]. Several oncogenic or tumor suppressor miRNAs have been shown to play diverse roles in the different cellular pathways of breast cancer development, such as cell proliferation, apoptosis, metastasis, and therapy resistance [10]. Analysis of miRNA expression profiles in Lebanese BC tissues revealed the dysregulation of several miRNAs in ER+ BC tissues as compared to normal adjacent ones. Of these miRNAs, miR-126 was significantly downregulated [12], with a fold change greater than 2 in young BC patients. miR-126 was reported to play diverse roles in different types of cancer, including breast cancer [13]. Hence, this stimulated our interest in further understanding its role in breast cancer development, specifically ER+ BC, which may ultimately serve as a therapeutic target.

First, we validated the downregulation of miR-126 revealed by the microarray analysis in ER+ BC tissues by RT-qPCR. Then, miR-126 was overexpressed in the ER+ BC cell line MCF-7 by transfection with miR-126 mimic to better understand its role. A significant decrease in cell proliferation and mammosphere-forming ability was observed in MCF-7 cells, with no effect on cell cycle progression. This shows that miR-126 might play a tumor-suppressive role in ER+ BC.

The downregulation of miR-126 that we have observed in the ER+ BC tissues is consistent with the results of a study that quantified a panel of miRNAs and showed that miR-126 is downregulated in BC tissues and their matched sera and showed a high correlation with the ER or PR expression levels, but showed no correlation with the age groups (<48 years and ≥48 years) [23].

In addition, multiple studies have shown that miR-126 overexpression decreased cell proliferation of MCF-7 [24,25,26,27], whereas another showed that it had no effect on the proliferation of this cell line [18]. Our results using MTT showed a significant decrease in the proliferation of MCF-7 72 h post transfection. Moreover, our data showed that overexpression of miR-126 had no effect on the cell cycle progression. However, two studies reported that miR-126 inhibited G0-G1 to S phase transition in MCF-7. These were conducted 24 h post transfection by DAPI and 60 h post transfection by PI staining [24,27]. As for mammosphere formation, ectopic expression of miR-126 resulted in a significant decrease in the mammosphere formation ability of MCF-7 when compared with that exhibited by NC-transfected cells.

Subsequently, we investigated the association of miR-126 with some potential targets that were determined by in silico analysis. Among the miR-126 potential targets that we studied is IRS1 which plays a crucial role in cell growth, primarily through the PI3K/Akt pathway [28]. IRS1 was validated as a direct target of miR-126 in BC by luciferase vector assay. Interestingly, they did not find any effect of miR-126 overexpression on the mRNA levels, while it decreased the protein levels [27]. This is in accordance with our data that revealed no significant change in IRS1 mRNA expression levels upon transfection with miR-126 mimic. The expression of SPRED1, a key player in VEGF signal transduction pathway that plays an important role in angiogenesis [29], was also investigated in our study. Cosan et al. reported that miR-126 mimic nonsignificantly increased the mRNA levels of SPRED1 in BC, which is consistent with our data [25]. They also showed that miR-126 inhibitor significantly increased SPRED1 mRNA levels. The inverse relation of PLK2 and CRK targets with miR-126 was not previously validated in BC, nor were they in our study.

Importantly, our data showed downregulation of the oncogenes SLC7A5 and PLXNB2 upon transfection with miR-126 mimic in MCF-7. Interestingly, our mRNA microarray analysis conducted on the Lebanese BC tissues previously mentioned revealed the upregulation of SLC7A5 and PLXNB2 that reiterates the tumor-suppressive role of miR-126 in suppressing these oncogenic targets.

PLXNB2 is a transmembrane receptor that plays a role in the development of the nervous system and cell migration [30]. Gurrapu et al. showed that upon knockdown of PLXNB2 or its ligand semaphoemaphoringrin 4C in different BC cell lines, growth was dramatically inhibited along with impairment of G2/M phase transition, cytokinesis defects, and cell senescence [31]. Xiang et al. showed that miR-126 overexpression decreased PLXNB2 mRNA and protein levels in ovarian cancer [32]. Similar to their results, we found that miR-126 overexpression in MCF-7 downregulated PLXNB2 mRNA levels.

SLC7A5 (LAT1), a member of the L-type amino acid transporter (LAT) family, is a sodium-independent transporter that transports large neutral amino acids such as leucine. SLC7A5 is overexpressed in several types of cancer and is related to cancer progression and aggressiveness [33]. Several studies have reported the high expression of SLC7A5 in the different BC subtypes (luminal A, luminal B, TNBC, and/or Her2+) and its role as a prognostic marker [34,35,36,37,38]. El Ansari et al. showed that high expression of SLC7A5 was associated with poor prognosis and poor survival outcome in the highly proliferative ER+ BC subtype (luminal B), indicating its role in the progression of the aggressive ER+ subtype and as a key therapeutic target [34]. These studies show that SLC7A5 might play a diagnostic and prognostic role in breast cancer. Two studies conducted by Shannan et al. showed that the system L (attributed by LAT1 and LAT2) is an important pathway for the uptake of essential neutral amino acids by BC cells (MCF-7 and MDA-MB-231) and, therefore, might play an important role in controlling cell growth [39,40]. Another study showed that the inhibition of SLC7A5 by BCH and other inhibitors inhibited the growth of the BC cell lines MCF-7, MDA-MB-231, and ZR-75-1 [41]. These studies highlight the role of SLC7A5 as a therapeutic target in breast cancer. As such, SLC7A5 might play an important role in the pathogenesis of breast cancer. Miko et al. showed that miR-126 overexpression suppressed SLC7A5 mRNA and protein levels in small cell lung cancer cells (SCLC) [42]. A recent study revealed that overexpression of SLC7A5 protein was significantly associated with histopathological grade in ER+ BC patients, and that SLC7A5 mRNA expression was positively correlated with the expression of marker of proliferation Ki-67 and hypoxia inducible factor 1 subunit alpha in ER+ BC patients [43]. Consistent with their results, we found that miR-126 overexpression in ER+ MCF-7 cell line downregulated SLC7A5 mRNA levels.

Finally, KM analysis was performed to determine the association between the expression levels of miR-126 or the potential targets, SLC7A5 and PLXNB2, with the overall survival of ER+ BC patients. High expression levels of miR-126 or low expression levels of SLC7A5 were associated with better overall survival, validating their role as potential prognostic biomarkers in ER+ breast cancer. It should be pointed out that miR-126 was shown to be elevated in formalin-fixed paraffin-embedded (FFPE) samples of indolent ductal in situ carcinoma (DCIS), but it was found to have low expression in high-risk DCIS and in invasive ductal carcinoma [44]. Another study showed that high levels of miR-126 in FFPE tissues of TNBC samples (86% are of early stage) was associated with a favorable TNBC outcome [45]. Similarly, the same results were reported in plasma of early BC (62% are ER+) as compared to metastatic BC [20]. Based on the role of miR-126 discussed in this manuscript and what is reported in the literature, it could act as a potential diagnostic and prognostic biomarker for early-stage BC.

## 4. Materials and Methods

### 4.1. Breast Cancer Tissue Specimens

To study miRNA expression levels in BC tissue samples, approval of the Institutional Review Board (IRB) at the American University of Beirut (AUB) and signed informed consents from the patients were obtained. Formalin-fixed paraffin-embedded (FFPE) sections were obtained from invasive ductal carcinoma specimens (*n* = 40) and normal adjacent tissues (NAT) (*n* = 19) identified by a pathologist at the American University of Beirut Medical Center (AUBMC) in Lebanon. All the BC tissue samples included in the study were of histotype invasive ductal carcinoma with no distant metastasis and with estrogen receptor and progesterone receptor-positive expression.

### 4.2. miRNA Microarray and Data Analysis

Twenty-four RNA samples (150 ng in 8 μL) isolated from young BC tissue samples were labeled using the FlashTag™ Biotin HSR Labeling Kit and later hybridized to the GeneChip miRNA 3.0 Array (Affymetrix Inc., Santa Clara, CA, USA) in accordance with the manufacturer’s instructions. Data were analyzed within the R statistical environment using Bioconductor (http://www.bioconductor.org (accessed on 6 January 2022)) packages, as mentioned previously [12].

### 4.3. Cell Culture

ER+ BC cell line MCF-7 was maintained in Dulbecco’s Modified Eagle’s Medium (DMEM) high glucose (Sigma Aldrich, Saint Louis, MO, USA) with 10% Fetal Bovine Serum (FBS) (Sigma Aldrich, USA), 1% sodium pyruvate, 1% penicillin/streptomycin, and 0.5% kanamycin. Cells were incubated at 37 °C with 5% CO_2_.

### 4.4. Cell Transfection

Cells were transfected with FAM-labeled miR-126 mimics and negative control duplex (NC) using Lipofectamine RNAiMAX Reagent (Invitrogen, Waltham, MA, USA) following the manufacturer’s instructions. Briefly, cells were seeded into 6-well plates or 96-well plates until 60–80% confluent and transiently transfected with 30 pmol or 5 pmol, respectively, with miR-126 mimics and NC. Then, cells were incubated at 37 °C for 24, 48, and/or 72 h prior to further analysis. Twenty-four hours post transfection, transfection efficiency was assessed by flow cytometry and RT-qPCR.

### 4.5. Total RNA Extraction

Total RNA was extracted from 40 tumor tissue and 19 NAT sections (for miR-126 expression) using RecoverAll^TM^ Total Nucleic Acid Isolation Kit for FFPE samples (Ambion, Austin, TX, USA) following the manufacturer’s instructions. Total RNA was extracted from the transfected cells using TRI Reagent (Sigma Aldrich, USA) according to the manufacturer’s instructions. Resulting RNA concentration and quality were assessed using DeNovix DS-11 FX spectrophotometer (Delware, DE, USA), and then stored at −80 °C. Only high-quality samples were used for downstream applications.

### 4.6. miRNA Expression by Quantitative Real-Time Polymerase Chain Reaction (RT-qPCR)

Reverse transcription of 10 ng of RNA was performed using TaqMan^®^ microRNA Reverse Transcription Kit (Applied Biosystems, Waltham, MA, USA). Multiplex cDNA master mixes were prepared, whereby miR-126 primers were used in each reaction with the endogenous control RNU6B which were part of the TaqMan^®^ microRNA Assays (Applied Biosystems, USA). RT-qPCR for miR-126 expression was performed using probes that are part of the TaqMan^®^ microRNA Assays and 2× TaqMan^®^ Universal Master Mix with no Amperase Uracil N-glycosylase (UNG) (Applied Biosystems, USA) on BioRad CFX96™ or CFX384™ Real-Time PCR Detection System (Hercules, CA, USA). The following steps were run: 10 min hold at 95 °C, 40 cycles of 15 s at 95 °C, and 60 s at 60 °C. miR-126 expression was normalized against the endogenous control RNU6B. Using the ΔΔCq, the relative expression of miR-126 was determined in the tumor samples compared to NAT and in the miR-126 mimic–transfected cells compared to the NC-transfected cells.

### 4.7. Flow Cytometry

Twenty-four hours post transfection, cells were analyzed on Guava EasyCyte8 Flow Cytometer (Millipore, Burlington, MA, USA) to determine the transfection efficiency. Briefly, cells were washed with 1× PBS; fluorescence intensity was adjusted upon loading the untransfected control (CTL) sample. Green Fluorescence (GRN-HLog) versus Side Scatter (FSC-HLin) was measured, and 10,000 events were collected. The percentages of transfected cells were quantitated by the software Guava Soft 2.7.

### 4.8. MTT Assay

In order to assess cell growth, cells were seeded into 96-well plates at a seeding density of 5000 cells/well and transfected as previously mentioned. Forty-eight and seventy-two hours post transfection, 10 μL of 5 mg/mL MTT (3-(4,5-dimethylthiazol-2-yl)-2,5-diphenyltetrazolium bromide) (Sigma Aldrich, USA) was added per well. Plates were incubated for 3 h, then, 100 μL of stop solution was added per well. Absorbance was measured at 595 nm by TriSTAR^2^ S LB 942 Multimode Reader (Berthold Technologies, Bad Wildbad, Germany) after incubation overnight.

### 4.9. Propidium Iodide Staining Assay

Cell cycle analysis was performed by propidium iodide (PI) staining. 500,000 cells of MCF-7 were seeded per well into 6-well plates and transfected as previously mentioned. Transfected cells were harvested 24 and 48 h post transfection. Briefly, cells were fixed in cold 70% ethanol, treated with 100 μL of 0.2 mg/mL Ribonuclease, then resuspended in 200 μL of 1× PBS + 20 μL of 1 mg/mL PI staining and incubated at 4 °C for 45 min protected from light. Samples were analyzed on Guava EasyCyte8 Flow Cytometer (Millipore, USA).

### 4.10. Mammospheres Assay

Forty-eight hours post transfection, cells were harvested and used to form spheres using MammoCult™ media (STEM CELL Technologies, Vancouver, BC, Canada) supplemented with 10% MammoCult™ Proliferation Supplement (STEM CELL Technologies, Canada), 0.2% heparin, and 0.5% hydrocortisone according to the manufacturer’s instructions. Afterwards, 40,000 cells/well of MCF-7 per condition were seeded in 2 mL complete MammoCult media in low adherent 6-well culture plates and incubated for 7 days. On day 7, spheres (>60 µm) were counted and mammosphere-forming efficiency (% MFE) was calculated as follows: % MFE = (number of mammospheres per well/number of seeded cells per well) × 100 [46]. To passage the spheres, 500 μL of pre-warmed Trypsin-EDTA was added for 1 min at 37 °C and the spheres were broken. Then, cells were seeded according to the previously mentioned densities. Carl Zeiss ZEN image software was used for the acquisition of bright field images of the mammospheres.

### 4.11. In Silico Predicted and Experimentally Validated Target Databases 

Two predicted target databases microT-CDS (Diana Tools, http://diana.imis.athena-innovation.gr/DianaTools/index.php (accessed on 6 January 2022)) and TargetScanHuman 7.2 (https://www.targetscan.org/vert_72/ (accessed on 6 January 2022)), and an experimentally validated database Tarbase 7.0 (Diana Tools) were utilized to search for predicted mRNA targets of hsa-miR-126-3p. A PubMed search was performed to check for validated miRNA–mRNA interaction in BC and other types of cancer, in addition, to checking the role of the resulting mRNA targets in BC.

### 4.12. Gene Expression of the Selected mRNA Targets by RT-qPCR

Reverse transcription of 1000 ng of RNA was performed using the iScript™ cDNA Synthesis Kit (Bio-Rad, Hercules, CA, USA). RT-qPCR for targets expression was performed using iTaq™ Universal SYBR Green^®^ Supermix (BioRad, Hercules, CA, USA) on BioRad CFX96™ or CFX384™ Real-Time PCR Detection System (Germany). The master mixes were prepared according to the selected concentration of each primer (Table 2). The following steps were run: 10 min hold at 94 °C, 40 cycles of 15 s at 94 °C, 60 s at 60 °C, as well as a melt curve 55 °C to 95 °C with an increment 0.5 °C for 0.05 s. mRNA expression was normalized against the housekeeping gene GAPDH. Using the ΔΔCq, the relative expression of the mRNA targets was determined in the miR-126 mimic–transfected cells compared to the NC-transfected cells.

### 4.13. In Silico Kaplan–Meier Analysis

miRpower web-tool was utilized to determine the survival of BC patients with dysregulated miR-126 or SLC7A5 expression. Metadata for Kaplan–Meier survival analysis were obtained using https://kmplot.com (accessed on 6 January 2022) [47]. The KM-plotter tool utilizes the METABRIC dataset which is deposited at the European Genome-Phenome Archive (http://www.ebi.ac.uk/ega/ (accessed on 31 January 2022)) and hosted by the European Bioinformatics Institute, under accession number EGAS00000000083. It contains RNA sequencing and whole genome sequencing for 2433 samples of primary breast tumors, of which 1262 samples had miRNA profiling data. Patients with ER+ status, with different nodal status and histological grade, were selected from the METABRIC dataset. This dataset was selected since it includes patients with a long-term follow up of 94.2 months. The median age of these patients is 61.5 (51.1–70.4). Kaplan–Meier plots were generated and a *p*-value < 0.05 was considered as a significant correlation between miRNA or target expression and survival.

### 4.14. Statistical Analysis

Statistical analysis was performed using GraphPad Prism 7 Software. Student’s *t*-test was used to analyze differences between the two groups. Data presented are the means ± SEM of three or four different experiments, as noted in the figure legends. A *p*-value < 0.05 was considered statistically significant (* *p* < 0.05, ** *p* < 0.01, *** *p* < 0.001).

## 5. Conclusions

In conclusion, miR-126 decreases proliferation and mammosphere formation of MCF-7, ER+ BC cell line, which suggests that this miR might play a tumor suppressor role in ER+ breast cancer through potentially targeting SLC7A5 (LAT1), a sodium-independent transporter overexpressed in cancer. This study also helps to shed light on the prognostic role of miR-126 and its potential target SLC7A5 in ER+ breast cancer that will need further validation in cohorts with larger sample sizes.

## Figures and Tables

**Figure 1 diagnostics-12-00745-f001:**
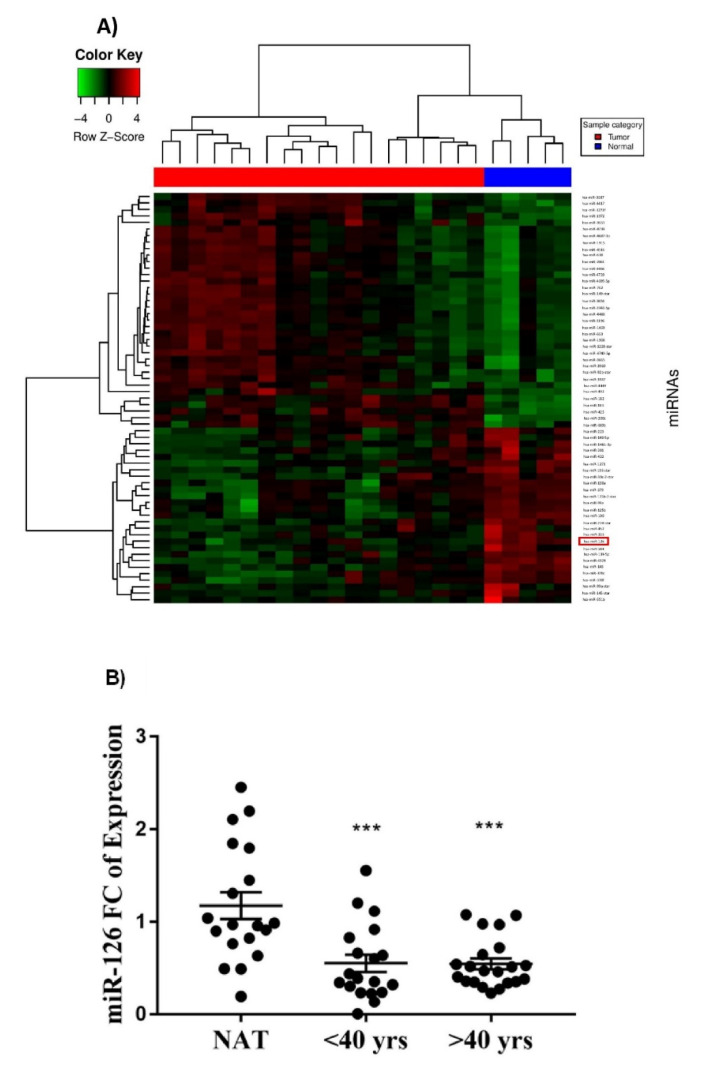
miR-126 expression in ER+ BC tissues versus NAT. (**A**) Heatmap of the differentially expressed miRNA in breast tumor tissue samples versus NAT taken from Lebanese ER+ BC patients < 40 years old (fold change > 2 and adjusted *p*-value < 0.05). Each row represents a differentially expressed miRNA and each column represents one of the 24 samples. The color key shown in the top left illustrates the relative expression level of an miRNA across all samples. (**B**) Dot plots show the fold change of miR-126 in 40 BC tissues normalized to the average of 19 NAT with RNU6B used as an endogenous control. Error bars represent SEM. *** denotes *p* < 0.001.

**Figure 2 diagnostics-12-00745-f002:**
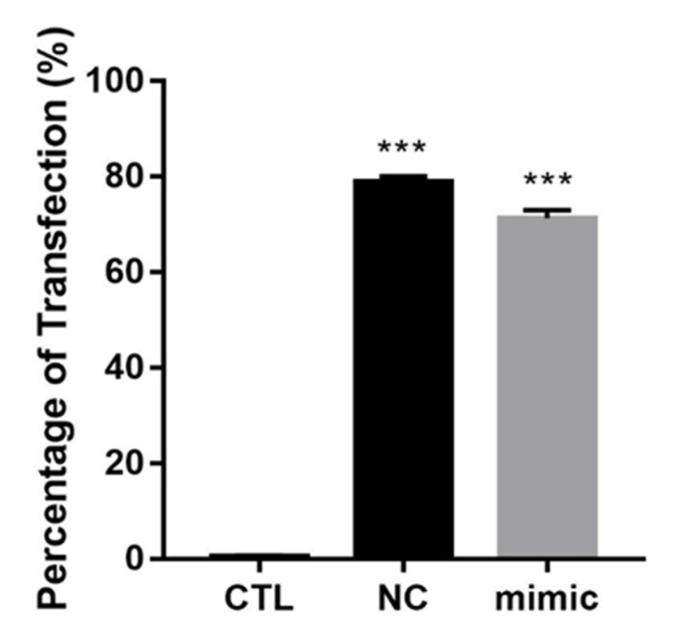
Transfection efficiency of miR-126 mimic in MCF-7. Percentage of transfection in FAM-labeled NC and miR-126 mimic–transfected cells as compared to untransfected control (CTL). Error bars represent SEM (*n* = 3). *** denotes *p* < 0.001.

**Figure 3 diagnostics-12-00745-f003:**
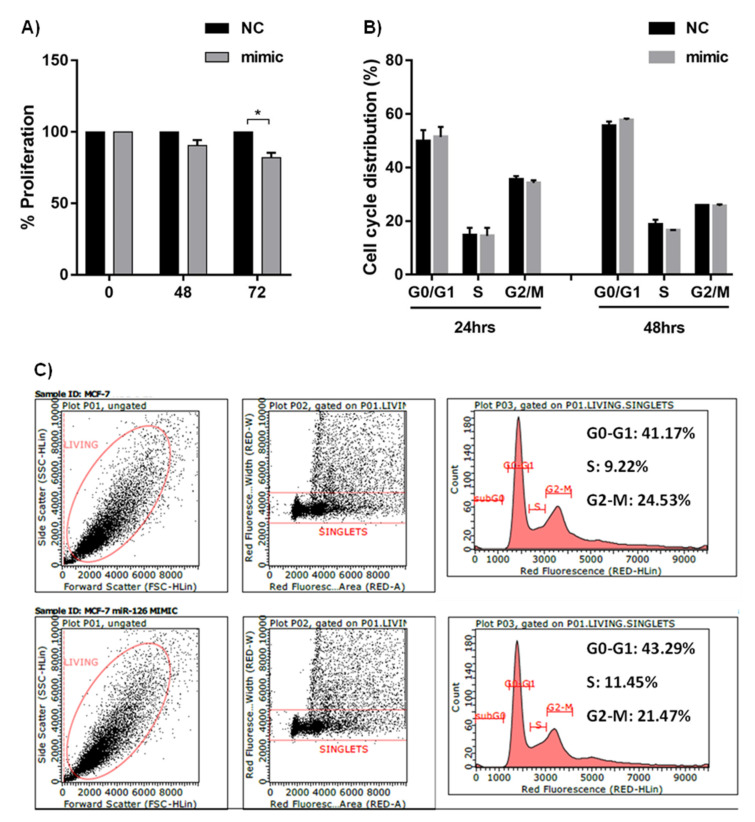
Effect of miR-126 on cell proliferation and cell cycle progression. (**A**): Proliferation of miR-126 mimic compared to NC-transfected cells by MTT assay at 48 and 72 h post transfection (*n* = 3). (**B**): Cell Cycle Analysis of the transfected cells by PI staining at 24 h (*n* = 3) and 48 h (*n* = 3) post transfection. (**C**): Representative figure of the flow cytometric analysis of the cell cycle 24 h post transfection. Error bars represent SEM. * denotes *p* < 0.05.

**Figure 4 diagnostics-12-00745-f004:**
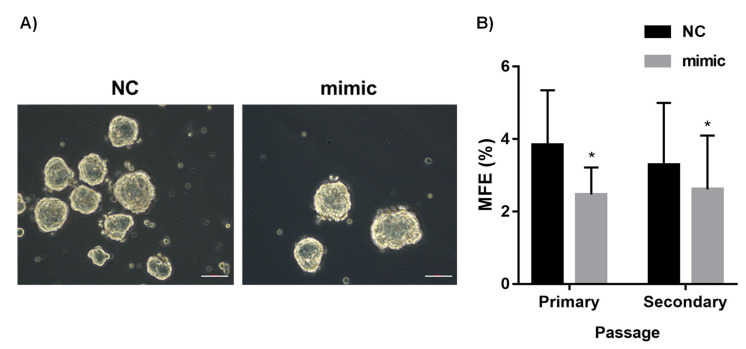
Mammosphere-forming ability of miR-126 mimic compared to NC-transfected cells. (**A**): Representative images of mammospheres formed by NC- and miR-126-transfected MCF-7. Scale bar: 100 µm. (**B**): Mammosphere-forming efficiency (MFE) of transfected MCF-7. Error bars represent SEM (*n* = 4). * denotes *p* < 0.05.

**Figure 5 diagnostics-12-00745-f005:**
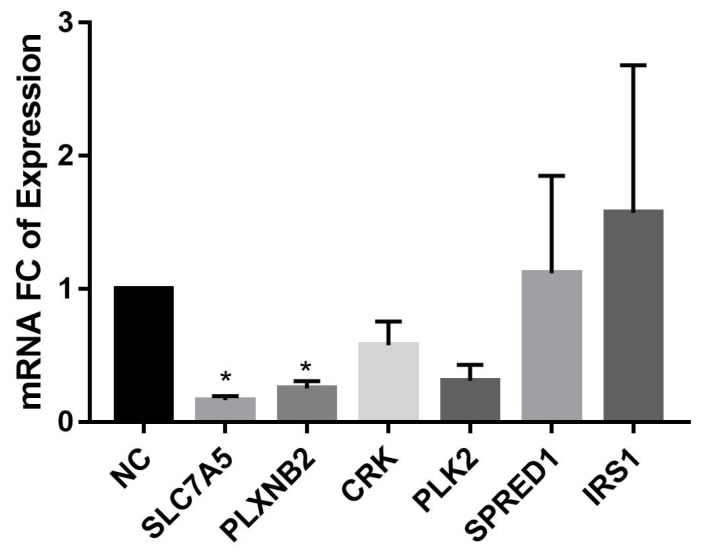
Expression levels of potential miR-126 targets in miR-126 mimic compared to NC-transfected MCF-7 24 h post transfection by RT-qPCR. GAPDH was used as an internal control. Error bars represent SEM (*n* = 3). * denotes *p* < 0.05.

**Figure 6 diagnostics-12-00745-f006:**
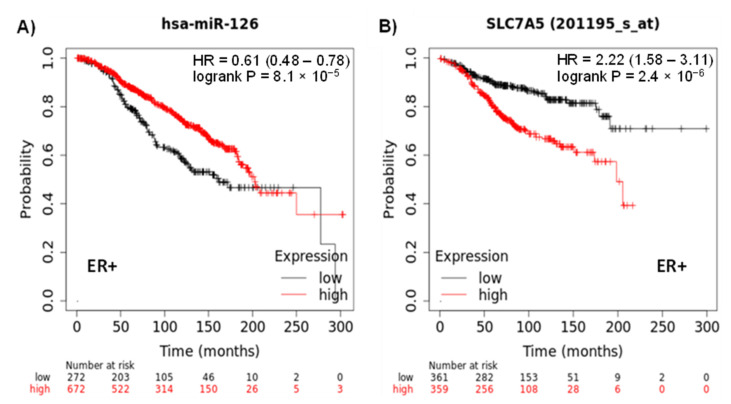
Correlation of the expression of miR-126 or SLC7A5 with overall survival (OS) of ER+ BC patients. In silico Kaplan–Meier plot of (**A**) hsa-miR-126 expression with OS of 944 ER+ BC patients, (**B**) SLC7A5 expression with OS of 720 ER+ BC patients. HR: hazard ratio.

**Table 1 diagnostics-12-00745-t001:** Selection Criteria of miR-126 Potential mRNA Targets in BC.

Target	Name	microT-CDS	TargetScan	Tarbase	Validated Relation of miRNA with BC	Validated Relation of miRNA with Other Cancers
SLC7A5	solute carrier family 7- member 5	Yes	Yes	Yes	No	Yes
PLXNB2	plexin B2	Yes	Yes	No	No	Yes
CRK	CRK proto-oncogene, adaptor protein	No	Yes	No	No	Yes
PLK2	polo-like kinase 2	Yes	Yes	No	No	Yes
SPRED1	Sprouty-related EVH1 domain containing 1	Yes	Yes	No	No	Yes
IRS1	insulin receptor substrate 1	Yes	Yes	No	Yes	Yes

**Table 2 diagnostics-12-00745-t002:** Sequences and Concentrations of Primers of GAPDH and miR-126 Predicted Targets Extracted from PrimerBank.

Gene	Primer	Sequence	Concentration (nM)
SLC7A5	SLC7A5-F	5′-GGAAGGGTGATGTGTCCAATC-3′	200
	SLC7A5-R	5′-TAATGCCAGCACAATGTTCCC-3′	
PLXNB2	PLXNB2-F	5′-AGCCTCTTCAAGGGCATCTG-3′	200
	PLXNB2-R	5′-GCCACGAAAGACTTCTCCCC-3′	
CRK	CRK-F	5′-GGAGACATCTTGAGAATCCGGG-3′	200
	CRK-R	5′-ACGTAAGGGACTGGAATCATCC-3′	
PLK2	PLK2-F	5′-CTACGCCGCAAAAATTATTCCTC-3′	200
	PLK2-R	5′-TCTTTGTCCTCGAAGTAGTGGT-3′	
SPRED1	SPRED1-F	5′-CAGCCAGGCTTGGACATTCA-3′	400
	SPRED1-R	5′-TGGGACTTTAGGCTTCCACAT-3′	
IRS1	IRS1-F	5′-CCCAGGACCCGCATTCAAA-3′	500
	IRS1-R	5′-GGCGGTAGATACCAATCAGGT-3′	
GAPDH	GAPDH-F	5′-ACAACTTTGGTATCGTGGAAGG-3′	500
	GAPDH-R	5′-GCCATCACGCCACAGTTTC-3′	

## Data Availability

The data presented in this study are available on request from the corresponding author.

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
