# Peer review of "miR-126 Decreases Proliferation and Mammosphere Formation of MCF-7 and Predicts Prognosis of ER+ Breast Cancer"

_diagnostics, 2022, doi:10.3390/diagnostics12030745_

Round 1

Reviewer 1 Report

1) The abstract needs to be reformed, as there are several unnecessary sentences (e.g., "Then, in silico analysis was done to determine the potential targets of the corresponding miRNA" could be written as "in silico analysis determined the potential miR-126 targets").

2) In the abstract, the authors state "RT-qPCR data showed that miR-126 overexpression significantly downregulated SLC7A5 and PLXNB2 mRNA levels in MCF-7". The only means to show that miR-126 downregulates its mRNA target is through in-vitro functional assays and detection of the gene product using Western blotting. A mere correlation between the expression of miR-126 and its putative dysregulated targets in miR-126 mimic transfected cells is of no real biological importance.

3) In the Introduction, after the sentence "...We reported the significant dysregulation of 173 mature miRNAs in ER+ BC tissues as compared to normal adjacent tissues 61 (NAT)", the authors should add the corresponding reference [12]; not in the next sentence.

4) The authors state that miR-126 was downregulated in <40 yrs old ER+ BCs compared to NAT. What about between ER+ BC <40 yrs old and >40 yrs old? In addition, the validation should focus on the 19 samples aged <40 years.

5) The HCl in Fig. 1A has a really poor resolution and miRNA names on the y-axis cannot be read. In addition, which samples are <40 yrs and which >40 yrs?

6) Is miR-126 endogenously expressed in MCF-7 cells? If yes, then they are not a good choice of a cell line to transfect with miR-129 mimics. Please elaborate. I don't understand what's the point in measuring miR-126 expression in MCF-7 cells transfected with the corresponding mimic. The results shown in Fig. 2B are neither surprising nor worth being presented.

7) Please provide the overall survival results for the ER- breast cancer patients expressing high and low miR-126 or SLC7A5, respectively.

Author Response

We are grateful for the thoughtful and critical comments.  We tried to provide our responses to the reviews we received, which we hope clarified and addressed the concerns raised by the reviewers.

1) The abstract needs to be reformed, as there are several unnecessary sentences (e.g., "Then, in silico analysis was done to determine the potential targets of the corresponding miRNA" could be written as "in silico analysis determined the potential miR-126 targets").

Thank you for your comment. The abstract was reformed as per your recommendation.

2) In the abstract, the authors state "RT-qPCR data showed that miR-126 overexpression significantly downregulated SLC7A5 and PLXNB2 mRNA levels in MCF-7". The only means to show that miR-126 downregulates its mRNA target is through in-vitro functional assays and detection of the gene product using Western blotting. A mere correlation between the expression of miR-126 and its putative dysregulated targets in miR-126 mimic transfected cells is of no real biological importance.

Yes sure we agree with the reviewer and that is why we only suggested correlation between the miRNA and the predicted targets and that the direct interaction should be validated.

3) In the Introduction, after the sentence "...We reported the significant dysregulation of 173 mature miRNAs in ER+ BC tissues as compared to normal adjacent tissues 61 (NAT)", the authors should add the corresponding reference [12]; not in the next sentence.

Thank you for your comment. The reference was corrected in the revised manuscript.

4) The authors state that miR-126 was downregulated in <40 yrs old ER+ BCs compared to NAT. What about between ER+ BC <40 yrs old and >40 yrs old? In addition, the validation should focus on the 19 samples aged <40 years.

When looking at microarray data, miR-126 had only a fold change greater than 2 in younger than 40 compared to NAT. Upon validation using real time PCR, miR-126 was downregulated was found in patients of all ages.

5) The HCl in Fig. 1A has a really poor resolution and miRNA names on the y-axis cannot be read. In addition, which samples are <40 yrs and which >40 yrs?

Thank you for pointing this out. Figure 1A which contains only samples < 40 years was modified in the revised manuscript.

6) Is miR-126 endogenously expressed in MCF-7 cells? If yes, then they are not a good choice of a cell line to transfect with miR-129 mimics. Please elaborate. I don't understand what's the point in measuring miR-126 expression in MCF-7 cells transfected with the corresponding mimic. The results shown in Fig. 2B are neither surprising nor worth being presented.

We used MCF-7 cell line as a model for ER+ breast cancer and although it expresses miR-126, it was a lower level than MDA-MB-231 the ER- BC cell line we tested. Regarding the point of measuring miR-126 expression in MCF-7 cells transfected with the corresponding mimic, it was done in order to make sure that our transfection was efficient. This can be moved to supplementary data if you recommend this.

7) Please provide the overall survival results for the ER- breast cancer patients expressing high and low miR-126 or SLC7A5, respectively.

 Thank you for your comment. OS of ER- BC patients were added to the revised manuscript. The figure shows that there is no significant difference in OS of ER- BC patients between high and low miR-126 or SLC7A5 expressing groups.

Reviewer 2 Report

In the article (diagnostics-1433600), the authors made substantial efforts to explore the role of miRNA-126 on the tumorigenesis of estrogen receptor positive breast cancer. Previously, the same group has reported that the expression of miRNA-126 is significantly downregulated in Lebanese BC patients via a global miRNA microarray analysis. In the present study, the authors extend their findings to explore the effect of overexpression of miRNA-126, by transfection with miRNA-126 mimic, on cell proliferation, cell cycle progression, mammosphere formation, and expression of potential targets. The results revealed that overexpression of miRNA-126 significantly reduced cell proliferation and mammosphere formation activity, while it showed no effect on cell cycle progression. Accordingly, the authors performed an in silico analysis aiming at identifying the potential targets for miRNA-126. The results revealed 6 target proteins (SLC7A5, PLXNB2, CRK, PLK2, SPRED1, and IRS1). Interestingly, PCR analysis showed that miRNA-126 overexpression significantly reduced the mRNA levels of SLC7A5 and PLXNB2. These results indicate that role miRNA-126 in tumor suppressor could be attributed to its effect to downregulate SLC7A5 level.

Overall, this is an interesting study which is well-designed, structured and performed. I would recommend the publication of this study after addressing the following concerns;

1- the intro part is not well presented and citation of relevant and recent studies is missing.

2- In the methods part, for all mentioned/performed experiments the authors have not cited any study?? which reflect that they have invented these methods. Please cite each performed experiments with the relevant/original studies.

3- Throughout the manuscript, I could not find any description for, how the authors calculated their sample size?

4- it is suggested that the authors include the figures of flow cytometry measurements in the main manuscript, as it is informative.

5- The authors should include the expression of SLC7A5, PLXNB2, CRK, PLK2, SPRED1, and IRS1 for the duplex transfected cells with GAPDH. Figure 5 does not present how the mRNA level of different protein has been changed or not.

6- please add better resolution for figure 6.

7- Indeed, I'm not sure about the reliability of estimation of OS. The authors should add a reference in line 155. I will keep this point for other reviewers to check.

8- please modify the conclusion part to be extended and more precious.

9- In the abstract/throughout the manuscript/title, I do not agree with the authors that the presented results indicate that ''SLC7A5 might be considered potential prognostic biomarkers in ER+ breast cancer''. Since the authors have not fully investigated what the effect of downregulation of SLC7A5, I would not high the tune/overestimate in this regard.

Author Response

In the article (diagnostics-1433600), the authors made substantial efforts to explore the role of miRNA-126 on the tumorigenesis of estrogen receptor positive breast cancer. Previously, the same group has reported that the expression of miRNA-126 is significantly downregulated in Lebanese BC patients via a global miRNA microarray analysis. In the present study, the authors extend their findings to explore the effect of overexpression of miRNA-126, by transfection with miRNA-126 mimic, on cell proliferation, cell cycle progression, mammosphere formation, and expression of potential targets. The results revealed that overexpression of miRNA-126 significantly reduced cell proliferation and mammosphere formation activity, while it showed no effect on cell cycle progression. Accordingly, the authors performed an in silico analysis aiming at identifying the potential targets for miRNA-126. The results revealed 6 target proteins (SLC7A5, PLXNB2, CRK, PLK2, SPRED1, and IRS1). Interestingly, PCR analysis showed that miRNA-126 overexpression significantly reduced the mRNA levels of SLC7A5 and PLXNB2. These results indicate that role miRNA-126 in tumor suppressor could be attributed to its effect to downregulate SLC7A5 level.

Overall, this is an interesting study that is well-designed, structured and performed. I would recommend the publication of this study after addressing the following concerns;

We are grateful for the thoughtful and critical comments.  We tried to provide our responses to the reviews we received, which we hope clarified and addressed the concerns raised by the reviewers.

1) the intro part is not well presented and citation of relevant and recent studies is missing.

Thank you for your comment. The introduction was reformed as per your suggestion

2) In the methods part, for all mentioned/performed experiments the authors have not cited any study?? which reflect that they have invented these methods. Please cite each performed experiments with the relevant/original studies.

Thank you for your comment. All of the experiments that are well explained in the methods section are well-established known experiments that we always use and report in our publications.

 3) Throughout the manuscript, I could not find any description for, how the authors calculated their sample size?

Thank you for your valuable comment. The sample size was not calculated. We were limited with the number of samples since it is based on data generated from the previous paper that we had.

 4) it is suggested that the authors include the figures of flow cytometry measurements in the main manuscript, as it is informative.

Thank you for your comment. Flow cytometric figures were added to the manuscript as per your recommendation

Kindly check the attached document for the figure

5) The authors should include the expression of SLC7A5, PLXNB2, CRK, PLK2, SPRED1, and IRS1 for the duplex transfected cells with GAPDH. Figure 5 does not present how the mRNA level of different protein has been changed or not.

The graph in Figure 5, represents the fold change of the mRNA targets in the miR-126 mimic transfected cells compared to the NC transfected cells using the ΔΔCq and after normalizing the mRNA expression against GAPDH. Results showed that both SLC7A5 and PLXNB2 were significantly downregulated upon miR-126 overexpression. Here below is the graph with delta Ct of each mRNA in miR-126 transfected cells (p-value of SLC7A5 is 0.0128 and ou PLXNB2 is 0.0223).

Kindly check attached document for the figure

6) please add better resolution for figure 6.

Thank you for the comment. A better resolution of the figure was added.

7) Indeed, I'm not sure about the reliability of estimation of OS. The authors should add a reference in line 155. I will keep this point for other reviewers to check.

Thank you for the comment. This is the only survival analysis available for microRNAs in the in silico database.

8) please modify the conclusion part to be extended and more precious.

Thank you for your comment. The conclusion was adjusted as suggested.

9) In the abstract/throughout the manuscript/title, I do not agree with the authors that the presented results indicate that ''SLC7A5 might be considered potential prognostic biomarkers in ER+ breast cancer''. Since the authors have not fully investigated what the effect of downregulation of SLC7A5, I would not high the tune/overestimate in this regard.

Apologies for the confusion. We have indeed used this phrase of considering SLC7A5 as a potential prognostic biomarker only based on the survival analysis of BC patients from the in silico tools that we have reported and that is independent of its function, that needs further validation.

Round 2

Reviewer 1 Report

1) The miRNA names on the y-axis of Fig 1A are still of poor visibility.

2) Please move Fig. 2B to suppl. data.

3) The same should go for the overall survival results for the ER- breast cancer patients expressing high and low miR-126 or SLC7A5, respectively

Author Response

Thank you again for the valuable comments that improved our paper. 

1) The miRNA names on the y-axis of Fig 1A are still of poor visibility.

We have uploaded a better quality Fig1A in the revised manuscript

2) Please move Fig. 2B to suppl. data.

As suggested, Fig. 2B is now moved to supplementary Figures as S1

3) The same should go for the overall survival results for the ER- breast cancer patients expressing high and low miR-126 or SLC7A5, respectively

We also moved Fig 6 B showing the the overall survival results for the ER- breast cancer patients expressing high and low miR-126 or SLC7A5 to supplementadry data as S2. We also added the figure showing the OS of the PLXNB2 high and the low expressing groups to S3 instead of data not shown. 

We finally revised the English language and style of the manuscript and corrected the minor errors 

All the changes are in track changes in the revised submitted manuscript.

Best regards,

Reviewer 2 Report

thanks to the authors for addressing all concerns that have been raised. The manuscript has been significantly modified. I would recommend the publication of this interesting study.

Author Response

Thank you for all your valuable comments that improved our manuscript. We now revised the English language and style of the manuscript and corrected the minor errors. All the edits are in track changes in the revised submitted manuscript.

This manuscript is a resubmission of an earlier submission. The following is a list of the peer review reports and author responses from that submission.

Round 1

Reviewer 1 Report

In this manuscript entitled “miR-126 decreases proliferation and mammosphere formation of MCF-7 and predicts Prognosis of ER+ Breast Cancer”, the authors reported the effects of mirna126 overexpression in MCF-7 proliferation and mammosphere formation and suggested this miRNA as a possible prognostic marker for ER+ breast cancer. There are some issues about this study:

-As previously mentioned, miRNA microarray analysis performed by the same group on 73 ER+ BC tissues (45 invasive ductal carcinoma specimens and 17 NAT) revealed that miR-126 was significantly downregulated in patients of all ages. Then, they confirmed that finding on 40 tumors and 19 NAT. However, it is not mentioned mir126 expression in ER- BC patients, but in vitro experiments were also performed on MDA-MB-231 triple-negative breast cancer.

-They also reported that the same in vitro experiments were also performed from others. What is the novelty?

- What is the basal expression of miRNA126 in MCF7 and MDA-MB-231? Are there any differences between MCF7 and MDA-MB-231 in terms of proliferation and mammosphere formation in basal conditions?

- To hypothesize a correlation between miRNA126 and SLC7A5 protein and its effect on cell proliferation and mammosphere formation, it would have been appropriate to perform an additional set of experiments.

- In fig 3, the reduced proliferation of transfected MCF7 is statistical significant at 72 hours, but cell cycle was only performed at 48 hours. Why? How did the authors explain this finding?

- In fig 4 mammospheres are more in mirna126-transfected MCF7 compared to control in contrast to the graphical representation. SEM values are very high to obtain a statistical significance.

Reviewer 2 Report

I find your manuscript very interesting and very useful in the field of breast cancer therapy and prognosis. Great work in statistical analysis and your english is adequate. You could improve grammatical errors and morphology changes according to instruction for authors. 

Reviewer 3 Report

The present study appears to be an extension of authors' prior work in reference 12, published in Sci Rep in 2017. It investigated the role of mir-126 in estrogen receptor positive breast cancer, and looked at the effect on proliferation, cell cycle and sphere formation. 

The present is quite disappointing in terms of novelty and the results presented are quite boring even though they appear to be sound.

First mir-126 does not appear to have an effect on cycle cycle progression (Figure 3), which makes me think why the authors choose to highlight this result in a main figure. 

Then mir-126 only has a very weak effect in decreasing mammosphere forming efficiency in MCF7 and MDAMB231 cells (the pvalues highlighted of 0.03 and 0.07 are not statistically significant) (Figure 4).  

Together, given the above uninteresting results, I am puzzled why the authors chose to investigate mir-126 in the first place. To support my suspicion that mir-126 is uninteresting to study, I further digged up the result of authors' Sci Rep paper in 2017. The authors said that mir-126 was motivated by the Sci Rep paper. But searching for mir-126 did not return any hit in that paper, and it is not among the 75 miRNAs that significantly changed between 45 BC and 17 normal breast tissues (when I searched the supplementary table). Given this fact, I don't think the present study is very well motivated.

Lastly in terms of novelty, the authors ignored the contribution of a prior highly similar work: Wang et al, Correlation and quantitation of microRNA aberrant expression in tissues and sera from patients with breast tumor, Gynecologic Oncology, Vol 119, Iss 3, Dec 2010, pp586-593. This prior paper not only investigated the role of mir-126, but also comprehensively examined 5 other microRNAs. It also arrived at the same result as author's paper, in terms of survival analysis in ER+ patients, and conclusion about mir-126 being a tumor suppressor gene. 

A few other questions:

  • The authors want to investigate mir-126 in ER+ BC, but the cell line used MD-MBA231 is a ER- cell line. Why is a ER- cell line used?
  • The connection between mir-126 and SLC7A5 is really unclear. What is the mechanism by which mir-126 affects SLC7A5 expression? By this I mean where is SLC7A5 expressed in the cell? What is the cellular mechanism and downstream effect of this mir-126 mediated perturbation? I feel that just showing the survival curves of mir-126 and SLC7A5 are not enough scientific contribution for a paper.